# "What influences how well householders living in previously flooded communities feel they are protected or could recover from future flooding?: Results of a survey"

Maureen Twiddy[1,2]*, Sam Ramsden[3]

1 Hull York Medical School, University of Hull, Hull, United Kingdom, 2 Institute of Clinical and Applied Health Research, University of Hull, Hull, United Kingdom, 3 Integrated Catchment Solutions Programme, University of Leeds, Leeds, United Kingdom

* m.twiddy@hull.ac.uk

**Data Availability Statement:** All relevant data are within the manuscript and its Supporting information files.

## Abstract

In England, the Environment Agency (EA) estimates that over 3 million properties in England are at risk of surface water flooding. Heavy and prolonged rainfall that drives surface water flooding is projected to increase in the future due to climate change. This paper presents a quantitative secondary analysis of a cross-sectional household flood survey in a disadvantaged city in England heavily impacted by surface water flooding in 2007 and at severe risk of flooding in the future. The aim of this study was to examine how previous experience of flooding, demographic factors, and behaviours impact on feelings of protection against flooding and perceived ability to recover from flooding. Survey data were collected from residents in Hull in northern England in 2018, in areas impacted by major floods in 2007 when over 8,600 households were flooded. Valid responses were received from 453 households, of whom 37.3% were flooded or flooding damaged their house (n = 169), 14.6% had been disrupted by flooding (n = 66), 9.3% had been exposed to flooding (n = 42) and 176 (38.9%) had not experienced flooding. Over 22% felt they had very low protection against flooding, and over 25% would make a very slow recovery if they were flooded. Associations were found between gender and both low levels of protection against flooding. Females were less likely to feel confident in their recovery from flooding than males (OR 0.551). The findings support a need to focus on women's perceptions of flood vulnerability and capacity to cope and recover from flooding in flood and disaster management policy and practice, including providing effective support before, during and after flooding.

## 1 Introduction

It is estimated that 90% of global natural disasters are water-related, with floods accounting for over half of these [1] (other water related disasters include: landslides, storms, heat waves, wildfires, extreme cold, and waterborne disease outbreaks). Between 2001 and 2018, floods were responsible for almost 100,000 deaths and USD 500 billion in economic losses [2], and

**Funding:** SR received an award from Living with Water to undertake the original Living with Water Project (no grant number). The funders have played no role in the study design, data collection and analysis, decision to publish or preparation of the manuscript.

**Competing interests:** No competing interests.

flooding is expected to increase in both frequency and severity due to global climate change [3, 4]. Brisley [5] describes flooding as 'one of the biggest risks facing the UK, and the UK Environment Agency (EA)[6] estimates that in England '*2.7m properties at risk of river and coastal flooding, 3m properties at risk of surface water flooding and some 660,000 properties at risk from all sources: river, coastal and surface water*'.

In England, Flood Risk Management Agencies (RMAs) are responsible for managing flood risk and are increasingly focused on building flood resilience in communities. The Environment Agency's [7] vision is: '*a nation ready for, and resilient to, flooding and coastal change—today, tomorrow and to the year 2100*' and the EA is providing £200m in funding to a range of projects under the Flood and Coastal Resilience Innovation Programme (FCRIP) [8].

Resilience is a term that term that can be used to describe the ability of households and communities to respond to shocks and stresses, such as flooding, by returning to normal, or even improving their circumstances [9, 10]. An individual's appraisal of a stressor, in this case flooding, is an essential predictor of coping efforts [11] and coping strategies affect resilience [12, 13], i.e. the ability to bounce back from negative emotional experiences and adapt to stressful experiences [14]. Spiegel et al [15] argue that there are two different ways of reviewing vulnerability and resilience in communities: (i) assessments based on pre-defined indicators and (ii) assessments based on perceptions within a community. This paper uses the latter approach to understand perceived vulnerability and resilience. Therefore, key to a community's resilience is how well individuals in that community appraise and cope with the impact of flooding, and understanding where the burden is felt is essential if we are to support individuals and communities better.

Evidence shows that in the 12 months after a flood, UK populations affected by flooding have higher levels of common mental health problems, such as depression and anxiety than the UK average [16], with women more vulnerable than men, and adults <65 years more likely than those >65 to report mental health issues [17]. Few UK studies examine longer term mental health impacts, but there is some evidence to show higher levels of mental health problems are experienced by those who have been flooded compared to those not flooded (e.g.[18, 19]). Therefore, different groups may view the impact of flooding quite differently.

This article aims to add to our understanding of the longerterm impact of flooding by analysing survey data assessing feelings of vulnerability to flooding and perceptions of flood resilience in Kingston-Upon-Hull (Hull). Hull is a city in the north of England with high levels of socio-economic disadvantage, and which has suffered from previous flood events and is very vulnerable to future flooding. Hull is particularly vulnerable to surface water flooding 'as it is largely below sea level and relies on a pumped drainage system with no natural ways of drainage'[20]. a In 2007, heavy rainfall caused severe surface level flooding in Hull resulting in major disruption, with 20,000 people reportedly affected by the flooding and 8,600 households flooded [21]. There was also smaller tidal flooding from the Humber Estuary in 2013 which impacted many businesses and households and closed major roads [22].

In 2018, in a collaboration between academia and a partnership of local Flood Risk Management agencies (RMAs): Living with Water (which includes Hull City Council, Yorkshire Water and the Environment Agency), a survey was conducted to understand communities' experiences of the 2007 flooding and how they felt about flooding now. This paper provides a secondary data analysis of the results of the 2018 survey (https://www.hull.ac.uk/editor-assets/docs/hull-household-flooding-survey-final-report.pdf).

The survey asked participants about their experiences of the large scale 2007 surface water floods and the smaller 2013 tidal flood. As part of understanding current feelings about flooding, the survey asked two proxy questions to understand personal feelings of vulnerability to flooding and its impacts:

*Q*: *How well is your house protected against flooding*?

*Q*: *How quickly do you think you would recover if your household was affected by flooding*?

This paper investigates whether previous experience of flooding, demographic indicators, and behavioural factors influence how well householders living in previously flooded communities feel they are protected or could recover from future flooding as indicators of vulnerability to flooding.

## 2 Methods

### 2.1 Study design and participants

This secondary analysis was performed on data collected in the 2018 survey, and examines data not previously reported. Ethical approval was obtained from the University of Hull, Living with Water Project (Project 716042: Living H20 Socio-Impact Assessment). Ref No. 20172018563 Date: 11/09/2018.

Data were collected between 25 September and 15 October 2018 via a door-to-door survey conducted in three council wards in Hull; Beverley and Newland, Derringham and North Carr, with an online option to widen recruitment beyond in-person survey hours. All participants were informed about the purpose of the study and their right to withdraw at any time. In-person participants provided verbal informed consent which was documented by the researcher. For online respondents, an introduction to the survey was provided and consent was implied if they chose to complete the survey. All respondents were adults, minors under the age of 18 were excluded. The survey used a purposive sampling approach, specifically targeting areas known to have been flooded in 2007. The survey asked participants about their experiences of the 2007 and 2013 floods. One survey was completed for each household surveyed (see supplemental data for survey questions).

Description of the areas (wards) sampled: Derringham was the most severely affected by the 2007 flooding out of the three wards. Beverley and Newland is the most ethnically diverse of the wards and also has a high population of students. North Carr is the most deprived of the three wards, however there are pockets of deprivation within all three wards with Hull being identified as the 4th most deprived local authority in England and Wales [23]. None of the three wards were particularly affected by the 2013 tidal floods as they are not adjacent to the Humber estuary, although respondents could have been disrupted or exposed to flooding or lived in these areas during the flood.

Surveys were sent to approximately 2,000 unique households of which 457 were returned, providing sufficient data for analysis (154 respondents answered online; 303 answered in person during door-to-door interviews). A response rate of 15% was obtained from target areas. A copy of the survey questionnaire is provided as a S1 File.

The survey focused on areas that were flooded in 2007 to capture the voices of people who had experienced flooding and had not previously shared their experiences [24]. Survey data was collected from households within the chosen areas. There was no strict targeted sampling to ensure a balance between flood-affected and non-flood affected households nor to obtain representative demographic characteristics—in line with approaches by Becker et al.[25], Paranjothy et al.[26] and Waite et al.[27].

Demographic information was collected including: household type, age group, gender, disability, ethnicity, residence status, and employment status. Other demographic information such as income, education and religion were not collected as it was felt that these questions could impact on engagement by adding a burden onto respondents. The survey team felt justified in this approach as there were very few respondents that began the survey but did not

complete it (including none online—this is recorded on the online software). For gender, we asked 'What is your gender'. The main answers were female (n = 247) and male (n = 192), 14 respondents did not answer the question or answered 'shouldn't matter', 'prefer not to say' or 'N/A' and these have been classified as missing for the analysis.

Survey respondents were asked about their exposure to the 2007 and 2013 floods and were categorized as: a) flooded or flooding damaged house, b) disrupted by flooding, c) exposed to flooding, d) or not flooded, based on respondents' answers. For this analysis, we combined responses for people who answered they were flooded in 2007 and/or 2013 (four respondents were only flooded in 2013). When grouping was unclear, data from free text questions were used to determine flooding status.

To understand how people had prepared for future floods we provided a tick list of actions households could take to prepare and respond to flooding (e.g. developed a flood plan, a flood kit, signed up for Environment Agency Flood warnings, joining a flood group); and property-level flood resilience (PFR) measures implemented in the home and garden (e.g. moving valuables to a safe place, raising electrical sockets, improving drainage from the property, protecting green spaces). For fifty cases we also used free text data to determine the household flood protection measures taken. Each measure was scored separately for the analysis, with one point for each different type of measure.

Two questions were used to understand feelings of vulnerability to flooding on 5-point Likert scales: 1) '*How well is your house protected against flooding*? (Rank from 1 Very Low Protection to 5 Very High Protection)' and 2) '*How quickly do you think you would recover if your household was affected by flooding*?' (Rank from 1 Very Slow Recovery to 5 Very Fast Recovery). Initially, we planned to ask respondents about feelings of 'flood resilience', however in the pilot phase we found that the term 'resilience' put residents off participating because the term was not used in everyday language in the local community.

## 2.2 Data analysis

Statistical data analysis was undertaken using SPSS version 27 [28]. Descriptive statistics were calculated for all respondents. As data were not normally distributed, (Shapiro-Wilks tests significant) non-parametric tests were used, and median/IQR reported. We excluded four cases who had not answered both the protection and recovery questions, leaving 453 cases. To assess the impact of demographic variables, previous flooding experience and behaviour on the two outcomes: 1) protection against flooding; and 2) recovery from flooding—we undertook a series of analyses using the following variables:

Demographic variables:

household type (1 = alone, 2 = family; 3 = with relatives; 4 = shared house)

age group (1 = 18–24; 2 = 25–34; 3 = 35–50; 4 = 51–64; 5-65-79; 6 = over 80 years)

employment status (1 = caring for relatives; 2 = employed; 3 = unemployed; 4 = retired; 5 = self-employed; 6 = student)

gender (1 = female; 2 = male)

property tenure (1 = owner; 2 = rent)

previous flooding experience: (1 = flooded, 2 = disrupted, 3 = exposed, 4 = not affected)

disabled (0 = no, 1 = yes).

Over 90% of respondents (n = 401) reported being White British, so ethnicity was recoded as (1 = White British, 2 = other White background; 3 = all other ethnicities).

Behavioural actions to prepare against flooding were separated into:

1. Actions to prepare and respond to future flooding (developing a flood plan, preparing a flood kit, signing up to flood warnings and joining a flood group), scored from 0 to 3, to indicate 0, 1, 2, or 3 or more activities undertaken;

2. Property-level flood resilience measures (PFR) (moving valuables to a safe place, raising electrical sockets, flood proof doors and windows) and external/ land measures (improving drainage, maintaining planting and greenery) (scored as 0/1to indicate no action vs action).

We used Mann-Whitney U and Kruskal-Wallis (non-parametric ANOVA) to test for univariate associations between each predictor and outcomes. Bonferroni corrections were applied to post-hoc tests.

1. **Mann-Whitney U tests** were performed to determine if perceptions of recovery and protection varied by gender, property tenure, or adoption of PFR measures.

2. **One-way Kruskal-Wallis tests** were conducted to determine if perceptions of recovery and protection varied between groups for the following predictors: household type, age group, employment status, actions to prepare and respond to flooding; flood experience.

Ordinal regression models with PLUM analysis were developed to examine the association of significant predictor variables on level of protection and recovery scores.

## 3 Results

### 3.1 Demographic information

The demographics of survey respondents were similar to the population of Hull at the time of the survey. Most respondents lived as part of a family unit (64%), the majority were female (56%). All age groups were represented, and 74% of the working age sample were employed or self-employed, compared to 70% for the population of Hull [29]. 21% of respondents (n = 95) answered they had a disability, compared to 23% of the working age population [30]. 9% stated they were from a 'white other' or ethnic minority background, compared to 10.3% across Hull [29]; 64% were owner occupiers, compared to Hull average of 49.5% [29] (see Table 1).

### 3.2 Behavioural factors

Overall, most people had undertaken few actions to prepare and respond to flooding, with 35.8% having taken no steps. However, 23% had taken at least of three of the seven possible actions listed. Likewise, the majority of respondents (72.2%) had not put in place any PFR measures (Table 2).

### 3.3 Protection against flooding

As shown in Table 3, half of respondents ranked their homes protection against flooding as low or very low, with only 17% ranking their protection as high or very high (mean 2.5 SD 1.1, median 2.00, IQR = 1, n = 447).

### 3.4 Recovery from flooding

As shown in Table 4, around half of respondents (52.4%) ranked their ability to recover from future flooding as slow or very slow (mean 2.45, SD 1.1, median = 2.00, n = 441), with less than 20% feeling they would recover quickly.

**Table 1. Demographic frequencies of respondents.**

|  | Frequency | % |
|---|---|---|
| **Household type** |  |  |
| Live alone | 114 | 25.2 |
| Family Unit | 292 | 64.6 |
| With relatives | 26 | 5.8 |
| Shared House | 20 | 4.4 |
| **Total** | **452** | **100** |
| **Age group** |  |  |
| 18 to 24 | 43 | 9.5 |
| 25 to 34 | 64 | 14.2 |
| 35 to 50 | 98 | 21.7 |
| 51 to 64 | 106 | 23.5 |
| 65 to 79 | 107 | 23.7 |
| 80 and over | 34 | 7.5 |
| **Total** | **452** | **100** |
| **Ethnicity** |  |  |
| White British | 401 | 90.7 |
| Other White background | 12 | 3.6 |
| White and Asian | 3 | 0.7 |
| Pakistani | 2 | 0.5 |
| Any other Asian background | 4 | 0.9 |
| Black, African, Caribbean background | 5 | 1.1 |
| Irish | 1 | 0.2 |
| White and Black Caribbean | 1 | 0.2 |
| Arab | 1 | 0.2 |
| Indian | 2 | 0.5 |
| Chinese | 3 | 0.7 |
| Any other mixed or multiple ethnic backgrounds | 3 | 0.7 |
| **Employment** |  |  |
| Caring for relatives | 7 | 1.6 |
| Employed | 181 | 40.5 |
| Not currently working | 49 | 11 |
| Retired | 157 | 35.1 |
| Self Employed | 33 | 7.4 |
| Student | 20 | 4.5 |
| **Total** | **447** | **100** |
| **Gender** |  |  |
| Female | 247 | 56.3 |
| Male | 192 | 43.7 |
| **Total** | **439** | **100** |
| **Own or rent property** |  |  |
| Own | 290 | 64.0 |
| Rent | 163 | 36.0 |
| **Total** | **453** | **100** |
| **Previous experience of flooding** |  |  |
| Flooded or flooding damaged their house | 169 | 37.3 |
| Disrupted by flooding | 66 | 14.6 |
| Exposed to flooding | 42 | 9.3 |

(*Continued*)

**Table 1.** (Continued)

|  | Frequency | % |
|---|---|---|
| Not experienced flooding | 176 | 38.9 |
| **Total** | **453** | **100** |

## 3.5 Univariate analysis: Predictors of confidence in protection against future flooding

Next, we examined whether demographic factors were associated with feeling confident about protection against future flooding. The following demographic variables were tested using Kruskal-Wallis one-way ANOVA tests: household type, age group, ethnicity, employment, actions to prepare and respond to flooding, previous experience of flooding. Mann-Whitney U-tests were used to test the relationship for property tenure, gender, disability, PFR measures (yes/no).

Household type, age group, disability, ethnicity, employment, tenure type, previous experience of flooding, and whether people had implemented PFR measures were not significantly associated with confidence in protection against future flooding. However, the following significant associations were found between feelings of protection against flooding and 1) actions taken to prepare and respond to flooding, and 2) gender.

*Actions taken to prepare and respond to flooding*: The results of the Kruskal-Wallis showed that those who had taken no measures felt they were significantly less protected against flooding compared to those who had taken one measure (U = -38.05, SE = 15.58, p = 0.07). The differences between the rank totals of 200.05 (no flood measures), 238.84 (one measure), 235.14 (two measures) and 237.1 (three or more measures) were significant, H (3, n = 447) = 9.146 p = 0.03. Post hoc Mann-Whitney U-tests using a Bonferroni adjusted alpha level of 0.008 (0.05/6) was used to compare all pairs of groups.

*Gender*: The results of the Mann-Whitney U tests show women had significantly lower confidence in their *protection against future flooding* (n = 247, median = 2, IQR = 4, mean = 2.36,

**Table 2.** Preparations for future flooding.

|  | Frequency | % |
|---|---|---|
| **Actions to prepare and respond to flood risk** |  |  |
| 0 | 162 | 35.8 |
| 1 | 109 | 24.1 |
| 2 | 78 | 17.2 |
| 3 to 5 | 104 | 23 |
| **Total** | **453** | **100** |
| **Number of PFR Measures** |  |  |
| 0 | 327 | 72.2 |
| 1 | 55 | 12.1 |
| 2 | 23 | 5.1 |
| 3 | 24 | 5.3 |
| 4 | 14 | 3.1 |
| 5 or more | 10 | 2.2 |
| **Total** | **453** | **100** |

**Table 3. Protection against flooding.**

|  |  | Frequency | % |
|---|---|---|---|
|  | Very low | 101 | 22.6 |
|  | Low | 125 | 28 |
|  | Medium | 144 | 32.2 |
|  | High | 52 | 11.6 |
|  | Very high | 25 | 5.6 |
| **Total** |  | **447** | **100** |

SD = 1.098) than men reported (n = 192, median = 3, IQR = 4, mean = 2.69, SD = 1.152); (U = 26980.5, n = 433, p = 0002).

## 3.6 Univariate analysis: Predictors of confidence in recovery from future flooding

We tested whether confidence in recovery from future flooding varied by key the following demographic variables: household type, age group, disability, ethnicity, employment, effects of previous flooding, number of general flood protection measures, gender and property tenure using Mann-Whitney U-tests and Kruskal-Wallis tests.

Household type, disability, ethnicity, employment, tenure, previous experience of flooding, actions to prepare and respond to flooding, and whether people had taken PFR measures, were not significantly associated with confidence in recovering from flooding. However, the following significant associations were found between confidence in recovery and 1) Age Group, and 2) Gender.

*Age group*: The results of the Kruskal-Wallis test showed that confidence in recovery from future flooding varied by age, with those in the 51 to 64 age group feeling less confidence than other groups. The differences between the rank totals of 247.49 (age 18–24), 214.64 (age 25–34), 216.92 (age 35–50), 188.97 (age 51–64), 236.15 (age 65–79) and 207.23 (aged over 80 years) were significant, H (5, n = 440) = 12.92 p = 0.02. Post hoc Mann-Whitney U tests, using Bonferroni correction found this difference was not statistically significant, but when the standard significance level of p = 0.05 was applied, we found those in the 51 to 64 years age group fared worse than those aged 65 to 79 years (U = -47.18, SE = 17.19, p = 0.006), those aged 25 to 35 years (U = 52.66, SE = 19.51, p = 0.009), or those aged 18 to 24 years (U = 58.51, SE = 22.28, p = 0.009).

*Gender*: The Mann-Whitney U tests indicated that women (n = 247, median = 2, IQR = 4, mean = 2.28, SD = 1.132) were more likely to believe their recovery from future flooding

**Table 4. Recovery from flooding.**

|  |  | Frequency | % |
|---|---|---|---|
|  | Very slow | 112 | 25.4 |
|  | Slow | 119 | 27 |
|  | Medium | 129 | 29.3 |
|  | Fast | 61 | 13.8 |
|  | Very fast | 20 | 4.5 |
| **Total** |  | **441** | **100** |

**Table 5. Parameter estimates for predictors of recovery from flooding.**

| Variable | Estimate | Std. error | Wald | df | Sig. | 95% CI | | 95%CI | | |
|---|---|---|---|---|---|---|---|---|---|---|
| | | | | | | Lower bound | Upper bound | Odds ratio | Lower bound | Upper bound |
| Recovery score = 1 | -1.13 | .341 | 10.954 | 1 | .001 | -1.797 | -.46 | .324 | .166 | .631 |
| Recovery score = 2 | .105 | .336 | .098 | 1 | .754 | -.554 | .765 | 1.111 | .575 | 2.148 |
| Recovery score = 3 | 1.51 | .345 | 19.099 | 1 | .000 | .833 | 2.187 | 4.525 | 2.299 | 8.906 |
| Recovery score = 4 | 3.06 | .396 | 60.099 | 1 | .000 | 2.291 | 3.841 | 21.46 | 9.885 | 46.59 |
| Gender = female | -.596 | .179 | 11.022 | 1 | .001 | -.948 | -.244 | .551 | .388 | .783 |
| Gender = male | .000 | | | 0 | | | | 1.000 | | |
| Age = 18–24 | .574 | .424 | 1.829 | 1 | .176 | -.258 | 1.405 | 1.774 | .773 | 4.07 |
| Age = 25–34 | .712 | .399 | 3.175 | 1 | .075 | -.71 | 1.494 | 2.037 | .931 | 4.456 |
| Age = 35–50 | .299 | .374 | .637 | 1 | .425 | -.435 | 1.032 | 1.348 | .647 | 2.808 |
| Age = 51–64 | -.145 | .371 | .153 | 1 | .696 | -.872 | .582 | .865 | .418 | 1.79 |
| Age = 65–79 | .53 | .372 | 2.028 | 1 | .154 | -.200 | 1.261 | 1.7 | .819 | 3.527 |
| Age = over 80 years | .000 | | | 0 | | | | 1.00 | | |

would be slower than men (n = 192, median = 3, IQR = 4, mean = 2.65, SD = 1.142), (U = 26524, p = 0.001).

## 3.7 Ordinal regression analysis to identify the most significant predictors

**3.7.1 Protection from flooding.** Univariate analyses indicate that undertaking flood preparation and response actions, and gender were significantly associated with beliefs about flood protection. These were entered into an ordinal regression model, but the model was not a good fit ($X^2$ = 19.64, p<0.001) and the assumption of proportional odds was not checked (p = 0.03), so the results are not reported.

**3.7.2 Recovery from flooding.** The univariate analysis identified that age group and gender were associated with confidence in recovery from flooding. The results of an ordinal regression analysis, (Table 5), show that when both gender and age group were entered in the model only gender is a significant predictor of recovery from flooding, with age no longer predictive of recovery score. The key results were:

Gender: Females are significantly less likely to feel confident in their recovery from flooding compared to males. The odds ratio of 0.551 indicates that females are 45% less likely to move up a category in recovery confidence compared to males (0.551–1)*100 = 44.9% (CI 95%, 0.388 to 0.783).

Age Groups: The age group 51–64 shows a negative estimate but is not statistically significant (p = 0.696). Other age groups show positive estimates, with the age group 25–34 being close to significance (p = 0.075). The data indicate a trend where younger people may feel more confident in their recovery, but these effects are not strong enough to be conclusive in this analysis (Exp (-.145) = 0.153 p = .075 NS), with the odds of people aged 51–64 being in a higher category 13.5% lower ((.865–1)*100 = 13.5% CI 95%.418 to 1.79).

## 4. Discussion

This study found strong associations between gender and both feelings of low levels of protection against flooding, and confidence in recovering from flooding, with women less likely to feel they were protected or could recover from flooding compared to men. Those who had taken action to prepare and respond to flooding were more confident than those who had taken no general actions to prepare and respond to flooding. Age was predictive of confidence

in recovery from flooding with people aged 51 to 64 significantly less likely to feel they can recover from future floods than those in other age groups. However, when both age and gender were entered into a model to predict recovery from flooding, gender was found to be the stronger predictor with age no longer predictive when gender was also considered.

The findings suggest that we need to focus on the perceived vulnerability of people aged 51 to 64 to slow flood recovery. This age group often have multiple responsibilities (property, family and caring responsibilities for children and grandchildren, work) and increasing health issues, but would not receive or qualify for the same level of support as people aged 65 and older who could have more ongoing connection with agencies [31, 32]. However, whether these factors account for our findings would need further investigation as existing studies focus on the needs of older people aged over 65 and identify they are more likely to have mobility constraints, suffer more consequences from flooding and require care and support during a flood [31, 33, 34].

Our results show that people who had taken any action to prepare or respond to flooding felt more protected than people who had taken no actions, which reinforces arguments for engaging people flood resilience programming and taking individual actions. However, a key finding of this study was the impact of gender on feelings of vulnerability to flooding, as we found that women feel less protected against future flooding and would have a slower recovery from flooding. Our findings reveal that living in an area affected by flooding in the past and at risk of future flooding leaves women feeling more vulnerable to flooding and its impacts than men. This indicates that there needs to be a specific focus on women in Flood Risk Management Agencies (RMA) programmes in the community to help them feel less vulnerable to flooding. The need to focus on women resonates strongly with a range of studies from both the global north and global south that argue that women are more vulnerable to the impacts of natural disasters than men [35–37]. Within the literature there is a strong focus on the unequal vulnerability of women to the impacts flooding, the need for greater protection from flooding, and support for better recovery from flooding as an essential platform for building flood resilience. Our findings support this conclusion and demonstrate that these feelings of vulnerability reported by women is related to gender, not just previous exposure to flooding.

UN Women [38] state that *'Gender inequality coupled with the climate crisis is one of the greatest challenges of our time. It poses threats to ways of life, livelihoods, health, safety and security for women and girls around the world'* and the joint Women's Environment and Development Organization (WEDO) and IUCN report [39] states that women and children are 14 times more likely to die than men during disasters. The experience of women during disasters in the global north could be very different than in the global south, although several studies outline the serious gender related impacts of Hurricane Katrina in the US, including negative impacts on care and parenting, mental health, gender-based violence [40, 41], and with strong influences of injustices in race and class also creating unequal impacts of flooding [42]. There are fewer studies researching gender aspects of flood vulnerability and resilience in the UK. However, Paranjothy et al.[26] investigated the mental health impacts of the 2007 floods in the UK context and found greater 'mental health impact for women, and for those with prior health problems' in the wider context of impacts being greater for those that experience flooding in the home. Sims et al [35] investigated the impacts of the 2007 floods in Hull and describes how many women have caring responsibilities which exacerbates the impacts of flooding, including increasing the stress of supporting the family to recover from flooding. Akerkar and Fordham [17] note that the UK government follows *'gender neutral policies in their disaster planning and management based upon a misconception that the gender gap has been eliminated'* and this approach appears to be reflected in RMA flood resilience programming in communities.

Our results add to the UK based research by showing an urgent need to consider gender and support women in community-level programmes to reduce vulnerability to flooding and

to build flood resilience and to also consider the impacts of age. This would require a shift in focus as many RMA flood resilience programmes aim to build household flood resilience through broad approaches focused on geographical communities, rather than targeting specific demographic groups. New approaches focusing on gender could include working with the third sector including charities and faith-based organisations that already work with women, and bring women together to develop flood plans, identify key actions and contacts, and simulate flood drills. During this specific work with women, researchers could conduct interviews or focus groups to identify specific concerns which would allow for more precise and actionable policy recommendations. This approach could also be expanded to ensure it covers people aged 50 and over.

In terms of flood policy, Paranjothy et al [26] make a strong recommendation to address the 'psychosocial and mental health impact of flooding' and that 'improved strategies for minimising disruption to essential services and financial worries need to be built into emergency preparedness and response systems'. There could also be a specific focus on the needs of women within these strategies, for instance around the links between caring responsibilities, the provision of essential services and financial support. At the time of this study in the UK, small grants are available to flooded households to improve properties (depending on criteria) [43], but these do not take account of income levels, family sizes and caring responsibilities. These activities should also be reinforced by a change in government policy to differentiate between men and women in disaster planning and response[17].

There is also a need to focus on women in flood vulnerability and resilience research. For instance, Masud et al [44] also argue that 'research approaches that focus on girls and women are needed with more focus on female groups to encourage a more just and inclusive disaster management approach'. Although some UK studies collect data from men and women on the impacts of flooding, very few disaggregate their findings to explore the experience of women during flooding [see [26, 35].

## Limitations and strengths

A limitation of the present study is its retrospective nature. Respondents were asked to recall the impact of flooding after 11 years, so this could have impacted on the relationship between previous experience of flooding and what people feel about flooding now. The survey did not aim to achieve a representative sample but did reach a sample broadly representative of the Hull population at the time. The use of a purposive sampling approach using in-person door-to-door interviews and online methods maximized opportunities to participate in a relatively short space of time (four weeks). Although the door-to-door interviews took place during working hours, the addition of online methods allowed working age residents to respond, and this is reflected in the demographics of respondents. The survey could be critiqued for not seeking information such as income and education, however the aim was to maximise engagement by reducing the burden on respondents in answering questions. Finally, there is a possibility of self-selection bias in such surveys, with those greatest affected most likely to reply. However, the large sample size, and the distribution of data indicate that this was a topic of interest to the wider community, as half of respondents had not personally experienced flooding, but wanted to share their views.

## 5 Conclusion

The floods occurred in 2007, and since there have been improvements made to the flood defenses throughout Hull. Despite this, women felt less protected from flooding than men and the felt they would take the longest to recover, so 'flood resilience programmes' still have much

to do to reduce feelings of vulnerability to flooding amongst different demographic groups. This paper identified a need to focus on women, while also arguing for further investigation of the impacts of age. The findings suggest a gendered approach is needed which focuses on supporting women at different stages of flooding, (preflood, during the flood, post flood recovery) both locally and at a policy level. This could include:

*Pre-flood and flood prevention*:

- Specific work with women who live in homes that have flooded before,

- A focus on practical actions and measures to help people: flood planning, ensuring access to good quality house insurance, clear information on where to go for help (including contacts names and numbers), and support for the implementation of PFR measures.

*During a flood*:

- A focus on supporting women with caring responsibilities, older women, women with heath requirements

*Post flood (recovery)*:

- Supporting women with caring responsibilities, older women and with health challenges. This should include ensuring access to health services, child-care support and financial support to help recovery.

There needs to be urgent consideration of gender differences in flood vulnerability and resilience policy and planning, reinforced by inclusive approaches to research. This requires a multi-sectoral approach involving flood risk management agencies, the third sector (including both charities and faith-based organisations), and academic research, together with a strong policy support from central government. For example, central government could provide more financial support to enable RMAs to work with the third sector to specifically support women. In the UK, this would require a shift in UK government policy to recognise the importance of gender in disaster planning and response[17].

## Supporting information

**S1 Dataset.**
(XLS)

**S1 File. Final hull LWW survey 2018.**
(PDF)

## Acknowledgments

Thanks go to the Living with Water Partnership (Yorkshire Water, Hull City Council, East Riding of Yorkshire Council, the Environment Agency and the University of Hull) who were partners in the original survey. Thanks too, to the residents of Hull for their input into this survey.

## Author Contributions

**Conceptualization:** Maureen Twiddy.

**Formal analysis:** Maureen Twiddy, Sam Ramsden.

**Funding acquisition:** Sam Ramsden.

**Investigation:** Sam Ramsden.

**Methodology:** Maureen Twiddy, Sam Ramsden.

**Project administration:** Sam Ramsden.

**Writing – original draft:** Sam Ramsden.

**Writing – review & editing:** Maureen Twiddy, Sam Ramsden.

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
