## [Decision Letter · Decision Letter 0]

8 Jan 2024

PONE-D-23-41282“What influences how well householders living in previously flooded communities feel they are protected or could recover from future flooding? results of a survey”PLOS ONE

Dear Dr. Twiddy,

Thank you for submitting your manuscript to PLOS ONE. After careful consideration, we feel that it has merit but does not fully meet PLOS ONE’s publication criteria as it currently stands. Therefore, we invite you to submit a revised version of the manuscript that addresses the points raised during the review process. We would recommend "major revision" for the submission. We agree with both reviewers, especially on the clarification of terminology and concepts, improvement of the abstract, literature review enhancement, and more details on statistical analysis

We look forward to receiving your revised manuscript.

Kind regards,

Sutee Anantsuksomsri

Academic Editor

PLOS ONE

Journal Requirements:

Additional Editor Comments:

I would recommend "major revision" for the submission. I agree with both reviewers, especially on the clarification of terminology and concepts, improvement of the abstract, literature review enhancement, and more details on statistical analysis.

Reviewers' comments:

Reviewer's Responses to Questions

**Comments to the Author**

1. Is the manuscript technically sound, and do the data support the conclusions?

Reviewer #1: Yes

Reviewer #2: Yes

2. Has the statistical analysis been performed appropriately and rigorously? 

Reviewer #1: Yes

Reviewer #2: Yes

3. Have the authors made all data underlying the findings in their manuscript fully available?

Reviewer #1: Yes

Reviewer #2: Yes

4. Is the manuscript presented in an intelligible fashion and written in standard English?

Reviewer #1: Yes

Reviewer #2: Yes

5. Review Comments to the Author

Reviewer #1: The author should consider the following additional comments:

1. The abstract should clearly state the objectives of the paper.

2. The reader may have questions regarding the accuracy of the statement that flooding impacts more people than other environmental disasters, as well as what is meant by "environmental disasters" in this context.

3. The manuscript should explore potential confounders for statistical analyses, such as religion, income, and ethnic group.

4. It would be beneficial to provide information on Hull's population demographic characteristics and the relationships between the population and sample demographics.

5. Table 5 needs better explanation as it is currently difficult to understand without accompanying text.

6. Consider rearranging the section titled "Strength and Limitations" to "Limitations and Strength." The aforementioned points can be discussed as limitations of the study.

Reviewer #2: Overall, I find the study to be promising, but there are a few critical areas that require attention and improvement before it can be considered for publication. Please find detailed feedback below:

Insufficient Literature Review and Conceptual Framework:

The manuscript lacks a comprehensive literature review, particularly in establishing a robust theoretical framework connecting demographic factors and the perceived ability to cope and recover from floods. It is crucial to provide a more detailed theoretical foundation to convincingly support the research objectives. I recommend incorporating relevant literature that directly addresses the linkage between demographic context/factors and flood resilience. This will strengthen the theoretical underpinning of your study and ensure that the research is aligned with its stated objectives.

Clarity in Terminology – Resilience vs. Perceived Capacity:

The manuscript often uses the term "resilience" to describe the outcomes related to the feeling of recovering and the ability to protect oneself from floods. However, resilience and perceived capacity are distinct concepts. It is advisable to refrain from categorizing the findings as contributing to "resilience" without a more nuanced exploration of these terms. Consider revising the terminology to accurately reflect the study's focus on the perceived capacity to cope and recover from floods.

Illustration of Hazards and Challenges for Different Demographic Groups:

The extent of hazards and challenges faced by different age groups and genders is not sufficiently illustrated in the current manuscript. To enhance the understanding of the difficulties experienced by various demographic groups, I recommend providing a more detailed analysis of the hazards encountered and their impact on different age groups and genders. This will contribute to a more nuanced and comprehensive interpretation of the study findings.

Enhanced Policy Recommendations:

The policy recommendations lack a direct and explicit link to the discussion. Instead of suggesting "specific work" or "practical action," consider providing concrete and targeted policy recommendations that align with the research findings. For instance, during the flood prevention phase, consider suggesting measures such as increasing the experience of female and elderly populations through flood drills or simulation exercises. Additionally, consider conducting short interviews to identify specific concerns among these demographics, allowing for more precise and actionable policy recommendations. Such detailed and context-specific suggestions will enhance the practical utility of your research, particularly for the city of Hull.

I appreciate the effort you have put into this research, and I believe that addressing these concerns will significantly strengthen the manuscript. I look forward to reviewing a revised version that incorporates these suggestions.

6. PLOS authors have the option to publish the peer review history of their article (what does this mean?). If published, this will include your full peer review and any attached files.

Reviewer #1: No

Reviewer #2: **Yes: **Yanin Chivakidakarn Huyakorn

---

## [Author Response · Author response to Decision Letter 0]

2 Apr 2024

We have responded fully to the reviewer comments and submitted our responses as a document with this resubmission. We thank the editor for the additional time to revise the manuscript. This has been helpful. 

We have uploaded documents as requested and ensured these meet the journal requirements.

---

## [Decision Letter · Decision Letter 1]

7 Jun 2024

PONE-D-23-41282R1“What influences how well householders living in previously flooded communities feel they are protected or could recover from future flooding? results of a survey”PLOS ONE

Dear Dr. Twiddy,

Thank you for submitting your manuscript to PLOS ONE. After careful consideration, we feel that it has merit but does not fully meet PLOS ONE’s publication criteria as it currently stands. Therefore, we invite you to submit a revised version of the manuscript that addresses the points raised during the review process.

We look forward to receiving your revised manuscript.

Kind regards,

Sutee Anantsuksomsri

Academic Editor

PLOS ONE

Journal Requirements:

Additional Editor Comments:

Thank you for submitting your revised manuscript, "What influences how well householders living in previously flooded communities feel they are protected or could recover from future flooding? results of a survey," to PLOS ONE.

After reviewing the comments and suggestions provided by the reviewers, I recommend a Minor Revision for your submission.

Overall, your revised manuscript is much improved. However, there are some crucial issues that need to be addressed. I agree with Reviewer 1 that the regression results and the conclusion need some more improvement.

Reviewers' comments:

Reviewer's Responses to Questions

**Comments to the Author**

1. If the authors have adequately addressed your comments raised in a previous round of review and you feel that this manuscript is now acceptable for publication, you may indicate that here to bypass the “Comments to the Author” section, enter your conflict of interest statement in the “Confidential to Editor” section, and submit your "Accept" recommendation.

Reviewer #1: All comments have been addressed

Reviewer #2: All comments have been addressed

2. Is the manuscript technically sound, and do the data support the conclusions?

Reviewer #1: Yes

Reviewer #2: Yes

3. Has the statistical analysis been performed appropriately and rigorously? 

Reviewer #1: Yes

Reviewer #2: I Don't Know

4. Have the authors made all data underlying the findings in their manuscript fully available?

Reviewer #1: Yes

Reviewer #2: Yes

5. Is the manuscript presented in an intelligible fashion and written in standard English?

Reviewer #1: Yes

Reviewer #2: Yes

6. Review Comments to the Author

Reviewer #1: The manuscript was significantly improved. The descriptive statistics are well shown. The results, discussions, and conclusion are very interesting. The reviewer suggests the following three points:

1. Overall, the manuscript can improve its readability to be more attractive to readers. The reviewer hopes the authors can examine and polish the manuscript from the audience's viewpoint.

2. Compared to other sections, the ordinal regression analysis part (3.6) in the aforementioned 1st point could benefit from improvement, especially for expressions. For instance, Table 5 can provide a general explanation, making it easier for readers to comprehend the author's interpretations. The reviewer provides an example for the interpretive expression part, but authors are not required to follow it; the authors can simply refer to it and consider the points presented.

- Gender: Females are significantly less likely to feel confident in their recovery from flooding compared to males. The odds ratio of 0.551 indicates that females are 45% less likely to move up a category in recovery confidence compared to males.

- Age Groups: The age group 51-64 shows a negative estimate but is not statistically significant (p=0.696). Other age groups show positive estimates, with the age group 25-34 being close to significance (p=0.075), suggesting a trend where younger people may feel more confident in their recovery, but these effects are not strong enough to be conclusive in this analysis.

3. The conclusion section still needs improvement in its writing style.

Reviewer #2: The policy recommendations could benefit from increased specificity; however, considering the primary objectives of the research, they are deemed acceptable.

7. PLOS authors have the option to publish the peer review history of their article (what does this mean?). If published, this will include your full peer review and any attached files.

Reviewer #1: No

Reviewer #2: No

---

## [Author Response · Author response to Decision Letter 1]

14 Jun 2024

Response to reviewers

We have completed the revisions requested by reviewer 1 and the editor. These are also detailed in the response to reviewers uploaded document. 

Reviewer 1 asked us to polish the manuscript. This is a rather general comment – we have reviewed the manuscript again and made some changes to sections of text for clarity.

Reviewer 1 asked that the ordinal regression results be rewritten to provide greater clarity in expression. We have used much of the material suggested by the reviewer and added this to the section. We have retained the statistical description as this was requested by a previous review. This way we hope to meet the needs of both the generalist and more statistically interested reader. 

Reviewer 1 asked that we look at the conclusion and improve the writing. We have revisited this section again and removed some repetition, and hope this meets their expectations. 

Reviewer 2 mentioned that the policy recommendations could benefit form increase specificity, but accepts these are acceptable. We have not revised these as given this is a secondary analysis we do not feel it appropriate to be more specific.

In addition, we have been through all the references to ensure that the links work for the electronic papers and checked that none of the papers cited have been retracted.

---

## [Editor Report · Decision Letter 2]

21 Jun 2024

“What influences how well householders living in previously flooded communities feel they are protected or could recover from future flooding? results of a survey”

PONE-D-23-41282R2

Dear Dr. Twiddy,

We’re pleased to inform you that your manuscript has been judged scientifically suitable for publication and will be formally accepted for publication once it meets all outstanding technical requirements.

Kind regards,

Sutee Anantsuksomsri

Academic Editor

PLOS ONE

Additional Editor Comments (optional):

I am delighted to say that your paper has been thoroughly revised and is now in good shape for publication.